# Reliability Assessment of a Vision-Based Dynamic Displacement Measurement System Using an Unmanned Aerial Vehicle

**DOI:** 10.3390/s23063232

**Published:** 2023-03-17

**Authors:** Hongjin Kim, Guyeon Kim

**Affiliations:** 1Department of Architectural Engineering, Kyungpook National University, Daegu 41566, Republic of Korea; 2Department of Architectural, Civil, Environmental, and Energy Engineering, Kyungpook National University, Daegu 41566, Republic of Korea

**Keywords:** VDMS using UAV, shaking table test, system identification, reliability assessment

## Abstract

In recent years, many studies have been conducted on the vision-based displacement measurement system using an unmanned aerial vehicle, which has been used in actual structure measurements. In this study, the dynamic measurement reliability of a vision-based displacement measurement system using an unmanned aerial vehicle was examined by measuring various vibrations with a frequency of 0 to 3 Hz and a displacement of 0 to 100 mm. Furthermore, free vibration was applied to model structures with one and two stories, and the response was measured to examine the accuracy of identifying structural dynamic characteristics. The vibration measurement results demonstrated that the vision-based displacement measurement system using an unmanned aerial vehicle has an average root mean square percentage error of 0.662% compared with the laser distance sensor in all experiments. However, the errors were relatively large in the displacement measurement of 10 mm or less regardless of the frequency. In the structure measurements, all sensors demonstrated the same mode frequency based on the accelerometer, and the damping ratios were extremely similar, except for the laser distance sensor measurement value of the two-story structure. Mode shape estimation was obtained and compared using the modal assurance criterion value compared with the accelerometer, and the values for the vision-based displacement measurement system using an unmanned aerial vehicle were close to 1. According to these results, the vision-based displacement measurement system using an unmanned aerial vehicle demonstrated results similar to those of conventional displacement sensors and can thus replace conventional displacement sensors.

## 1. Introduction

The development of large structures, including ultra-high-rise structures and super-long spanning bridges, is becoming increasingly common. Thus, the importance of structural health monitoring (SHM) becomes more relevant. In SHM, basic responses (e.g., acceleration, displacement, deformation, and tilt) are used because they can clearly reflect the local and global behaviors of existing structures under various load conditions [1,2]. Among such basic responses, displacement measurement is an important method that directly exhibits high accuracy in low-frequency regions and the deformation geometry of a structure. However, there are limitations in measuring the displacement of an actual structure [3].

Sensors for displacement measurement can be classified into contact and noncontact sensors. Linear variable differential transducers (LVDTs) are representative contact sensors used for measuring longitudinal displacement. When a current flows in the coil inside such a sensor and the core is moved by the occurrence of displacement, an induced current is generated in the auxiliary coil, which is converted into distance. The sensor has a long service life because of slight friction, and it exhibits high accuracy because a corresponding output is generated even for the slightest movement of the core. However, applying such a sensor to an actual structure is difficult because it must be in direct contact with the structure for displacement measurement and should be installed at a fixed point independently from the structure [4].

Alternatively, noncontact sensors include laser distance sensors (LDSs) and the global positioning system (GPS). LDSs measure displacement by irradiating a laser beam at the point to be measured and determining the time required by the beam to reflect and return. Unlike LVDTs, they can easily ensure high data reliability because of performance improvements in sensors. However, applying noncontact sensors to actual structures is difficult because the effective distance is short, and they must be installed independently from structures. Furthermore, the use of the GPS for measuring dynamic displacement is challenging because it is costly, has a small amount of measurement data per unit time, and has a relatively wide error range [5,6,7,8].

Computer vision detection has been utilized in numerous studies for various purposes, including crack detection, photogrammetry, damage detection, and VDMSs [9,10,11,12,13]. By using deep learning techniques, cracks and damages can be accurately detected, and this information can be used as a maintenance technique for structures [9,10]. Moreover, a study has been conducted to estimate the deformation of a structure by generating a 3D point cloud from an image using a photogrammetry technique [11,12]. In particular, Tang et al. applied a mark-free vision measurement system to the seismic performance evaluation of a reclaimed aggregate concrete filled steel pipe column and generated a hysteresis curve [13]. The results demonstrated good agreement with the theoretical curve.

Thus, a vision-based displacement measurement system (VDMS) has been proposed for alternative sensors. A VDMS is easy to install and dismantle because the measurement equipment, which uses a hand-held camcorder, is relatively inexpensive and highly portable [14]. It has fewer restrictions under field conditions compared with conventional sensors because it requires no separate equipment other than the hand-held camcorder during measurement and performs noncontact remote measurements.

Research on vision-based displacement measurement commenced in the mid-1980s, and there have been significant advances in computer vision algorithms. For instance, Peters et al., for the first time, developed a vision software program that can measure displacement and strain in an experiment [15]. However, the initial VDMS could not avoid aliasing in actual civil engineering structures and buildings because cameras then had a small number of frames per second and would not capture small displacements of structures because of their low resolutions [16,17].

Thereafter, owing to the development of cameras and algorithms, displacement measurements at actual buildings and static displacement measurements and small-scale verification tests in laboratories have been performed. For instance, Part et al. developed a three-dimensional (3D) displacement measurement model using a motion capture system. They measured the free vibration of a small three-story structure and compared the results with those obtained using an LDS [18]. Meanwhile, Choi et al. applied an algorithm to adjust the size of the region of interest (ROI) and ROI updating to the VDMS to improve the accuracy of measurements from digital images [19]. Kim et al. performed measurements of laboratory-scale small structures and actual structures using the VDMS and verified the validity of the obtained dynamic characteristics [20]. Additionally, various studies have been conducted to develop vision systems for real-time vision-based displacement measurements and an actual structure [21,22,23]. Lydon et al. developed a displacement measurement system using a commercial action camera equipped with a low-cost telescopic lens and experimentally verified its performance in a laboratory. Based on the obtained results, they measured the displacement response on two actual bridges and verified the effectiveness of the system [24]. Given that VDMS exhibits low measurement accuracy as the measurement distance increases because of the reduction in the image resolution from the target, its application to high-rise buildings and large structures, which require long-distance measurements, is difficult.

Vision-based monitoring technologies encompass a variety of tools, including video recorders, virtual vision sensors, digital image correlation, and 3D laser Doppler vibrometry [25,26]. Scislo proposed an innovative solution to reduce the time required for quality control by utilizing a 3D laser Doppler vibration measurement system. This system allows for the measurement of natural frequencies, damping ratios, and mode shapes of civil structures or building structural elements, making it a valuable tool for modal response analysis. For example, a study conducted by used a laser Doppler vibrometer to compare the modal response of a manufactured product with an ideal template [26].

Recently, various studies have been conducted for monitoring civil engineering facilities to detect static deformation, local cracks, and corrosion using unmanned aerial vehicles (UAVs) [27,28,29,30,31]. In particular, studies have been conducted on VDMS using UAVs. The use of UAVs can increase measurement accuracy because target structures can be accessed and high-resolution images can be collected regardless of the measurement target. UAVs move and collect images using the abnormal camera shake in six degrees of freedom (DOF) in real time, unlike the existing VDMS. However, a correction method is required, and studies have been conducted to calculate the movement of UAVs by tracking fixed points inside images and measuring the final behavior of a structure [32,33]. In this regard, Yoon et al. measured the dynamic displacement of a structure by estimating the movement of a UAV using a camera calibration technique based on the optical flow [34,35,36]. Moreover, Yan et al. and Weng et al. proposed a UAV-based vibration measurement method that combines a convolutional neural network (CNN) and the Kanade–Lucas–Tomasi optical flow methods [37,38]. This method uses the CNN to extract fixed points inside UAV images and track the movement of the target and background, and it does not require fixed points (e.g., markers) outside the structure. Additionally, Wang et al. conducted a shaking table test on a full-scale structure and measured its displacement using a VDMS using a UAV and a light and ranging detection scanner [39]. Han et al. measured the displacement of a structure via continuous scaling factor updating by generating a laser spot on the same surface as the target measurement point [40].

As described earlier, various studies have been conducted on the VDMS using UAVs, but a reliable measurement range must be clearly determined for its application in the field. In previous studies, the reliability and measurable range of the VDMS using UAVs were not considered. Hence, in this study, measurements were performed while the frequency of sine waves and displacement were varied, and the results were compared with those obtained using conventional displacement sensors in an experiment examining the dynamic measurement reliability of VDMSs using UAVs. As regards the method for correcting the movement of UAVs, the camera calibration technique based on a checkerboard was applied. Additionally, free vibration was generated in a model structure with one and two stories, and the accuracy for estimating the mode frequency, damping ratio, and mode shape was examined to verify the validity of the obtained dynamic characteristics in a structure.

## 2. Materials and Methods

### 2.1. Camera Calibration

The camera calibration technique is the process of obtaining a transformation matrix where a 3D object is projected onto a 2D image plane by acquiring transformation data (e.g., size, shape, rotation, and distortion) lost in the process of projecting a 3D object and space onto a 2D image. A 3D object is affected by the factors inside a camera, such as the lens, focal distance, and image sensor, during its output as an image. Therefore, the internal factors of the camera should be determined for the 3D restoration of the object within the image. Furthermore, the 3D positional relationship between the camera and the object should be considered. Accordingly, Figure 1 shows the 2D image coordinates and 3D world coordinates.

The transformation matrix between a point in 3D space X and the corresponding point in the image plane coordinate system of the 2D camera x is as follows [41].
(1)$$X={\left(X,Y,Z\right)}^{T}$$
(2)$$x={\left(x,y\right)}^{T}$$
(3)$$s\left[\begin{array}{c} x\\ 1 \end{array}\right]=\left[\begin{array}{ccc} f_{x} & S_{k} & c_{x}\\ 0 & f_{y} & c_{y}\\ 0 & 0 & 1 \end{array}\right]\left[\begin{array}{cccc} r_{11} & r_{12} & r_{13} & t_{1}\\ r_{21} & r_{22} & r_{23} & t_{2}\\ r_{31} & r_{32} & r_{33} & t_{3} \end{array}\right]\left[\begin{array}{c} X\\ 1 \end{array}\right]$$
where s denotes the scale factor, $f_{x}$ and $f_{y}$ denote the focal distances of the camera, and $c_{x}$ and $c_{y}$ denote the principal point, which is the center of the camera lens. Furthermore, $S_{k}$ denotes the skewness coefficient, $r_{ij}$ (i = 1,2,3, j = 1,2,3) denotes the rotational movement element, and $t_{i}$ (i = 1,2,3) denotes the parallel movement element. The first 3 × 3 matrix refers to the internal parameters of the camera, including the camera focal distance and angle between the image sensor and lens, whereas the second 3 × 4 matrix refers to external parameters, including geometric factors such as the camera position and direction.

Because projecting a 2D image to 3D world coordinates using a single camera has scale ambiguity, the Z coordinates of all moving targets remain constant during tests. Z in Equation (1) can be replaced by a constant value $\tilde{Z}$ to reconstruct the world coordinates of the target. Therefore, Equation (3) can be rewritten as Equation (4) [39].
(4)$$s\left[\begin{array}{c} x\\ 1 \end{array}\right]=\left[\begin{array}{ccc} f_{x} & S_{k} & c_{x}\\ 0 & f_{y} & c_{y}\\ 0 & 0 & 1 \end{array}\right]\left[\begin{array}{ccc} r_{11} & r_{12} & r_{13}\\ r_{21} & r_{22} & r_{23}\\ r_{31} & r_{32} & r_{33} \end{array}\right]\left[\begin{array}{c} X\\ Y\\ \tilde{Z} \end{array}\right]+\left[\begin{array}{ccc} f_{x} & S_{k} & c_{x}\\ 0 & f_{y} & c_{y}\\ 0 & 0 & 1 \end{array}\right]\left[\begin{array}{c} t_{1}\\ t_{2}\\ t_{3} \end{array}\right]$$

Equation (4) is rewritten as Equation (5) to calculate the motion of the target in global coordinates.
(5)$$\left[\begin{array}{c} X\\ Y\\ \tilde{Z} \end{array}\right]=s{\left[\begin{array}{ccc} r_{11} & r_{12} & r_{13}\\ r_{21} & r_{22} & r_{23}\\ r_{31} & r_{32} & r_{33} \end{array}\right]}^{-1}{\left[\begin{array}{ccc} f_{x} & S_{k} & c_{x}\\ 0 & f_{y} & c_{y}\\ 0 & 0 & 1 \end{array}\right]}^{-1}\left[\begin{array}{c} x\\ 1 \end{array}\right]-{\left[\begin{array}{ccc} r_{11} & r_{12} & r_{13}\\ r_{21} & r_{22} & r_{23}\\ r_{31} & r_{32} & r_{33} \end{array}\right]}^{-1}\left[\begin{array}{c} t_{1}\\ t_{2}\\ t_{3} \end{array}\right]$$

### 2.2. UAV Movement Correction

Unlike the existing VDMS technique, which measures displacement by attaching a checkerboard to the target structure to be measured, the rotation angle of the camera must be extracted using an additional fixed point at which displacement does not occur in the image. One checkerboard is attached to the structure that moves, whereas the other checkerboard is attached to the fixed point with no displacement. In this case, because locating the two checkerboards on the same plane is difficult, the distance between them must be considered to remove the displacement generated by camera rotation. In this study, the position of the calibration board was obtained using MATLAB 2022b’s Camera Calibration application [42].

The movement and rotation of the UAV must be eliminated to calibrate the actual displacement of the target structure. Figure 2 shows the displacement of the target structure, including the UAV’s movement and rotation and the displacement of the background target installed to calibrate the UAV’s movement and rotation. If the total displacement of the structure captured by the moving and rotating the UAV is assumed as $D_{1}$, then $D_{1}$ includes the actual displacement generated in the structure ($D_{r}$), the displacement generated by the UAV’s rotation ($D_{2}$), and the displacement generated by the UAV’s movement ($D_{3}$). Moreover, the total displacement of the background target is $D_{4}$, including the displacement generated by the UAV’s rotation at the background target ($D_{5}$) and the displacement generated by the UAV’s movement ($D_{3}$). Therefore, to measure the accurate displacement of the structure, the displacement generated by the UAV’s movement and rotation must be removed from the total displacement of the target structure in the following process.
(6)$$D_{5}=H_{2}\tan\theta$$
(7)$$D_{3}=D_{4}-D_{5}$$
(8)$$D_{5}-D_{2}=\left(H_{2}-H_{1}\right)\tan\theta$$
(9)$$D_{2}=D_{5}-\left(H_{2}-H_{1}\right)\tan\theta$$
(10)$$D_{r}=D_{1}-D_{2}-D_{3}$$
where $H_{1}$ denotes the distance between the camera and structure and $H_{2}$ denotes the distance between the UAV and the background target attached to the fixed point with no displacement. Note that θ denotes the rotation angle of the UAV with respect to the *Y*-axis.

## 3. Vibration Measurement Performance Verification of VDMS Using a UAV

### 3.1. Test Overview

In this study, a shaking table test was performed by applying sine waves while increasing the frequency by 0.2 Hz (from 0.2 to 3 Hz) and the displacement by 5 mm (from 5 to 100 mm) to examine the vibration measurement performance of the VDMS using a UAV. The measurements of the VDMS using a UAV were compared with those of conventional displacement measurement sensors (i.e., LVDT and LDS). UAV imaging was performed at a distance of ~5 m from the target, and the altitude was set to 1 m to capture the images of the target from the front. A grid-type checkerboard was used as the target for the VDMS using a UAV. The checkerboard had 12 edges, and the gap between the edges was 70 mm. The test time was set to be more than ten times the excitation period to secure sufficient data for the estimation of dynamic characteristics. Figure 3 shows the shaking table, background target, displacement sensor, and the scene of the experiment where the UAV captured the images of the target.

The measurement accuracy was evaluated by determining the root mean square (RMS). Equation (11) shows the calculation process, where $x_{i}$ is ith displacement data and n is the total number of displacement data. Furthermore, the natural frequency and magnitude errors were compared in the frequency domain.
(11)$$\mathrm{RMS}=\sqrt{\frac{\sum_{i=1}^{n}x_{i}{}^{2}}{n}}$$

### 3.2. Measurement Equipment Configuration

The experimental setup was composed of a shaking table that can apply sine waves, displacement sensors (LVDT and LDS), UAV for imaging, and a measurement target. A portable one-axis vibration device was used as the shaking table. It comprised a shaking table, a control panel, and a touch panel. When the frequency and displacement are set and an operational command is provided using the touch panel, three-phase signals for driving the motor are produced from the servo pack installed in the control panel. The shaking table was composed of Al of dimensions 550 mm × 350 mm × 75 mm. M5 tapped holes were drilled in both directions at 25 mm intervals on the upper table. KL3-W400 from KAIS, a Korean company [43], was used as the laser sensor, and DP-500E from Tokyo Sokki, a Japanese company, was used as the LVDT [44]. Measurements using the LDS and LVDT were simultaneously acquired using TMR-311 [45], a data logger from Japan that can adjust the sampling rate at regular intervals, and a sampling rate of 1000 Hz was set in the experiment for dynamic displacement measurement. Chinese company Dji’s MAVIC 3 [46] was used as a UAV, and the imaging resolution was set to 4K (3840 × 2160) to increase the accuracy of recognizing the edges of the target. The frame rate was set to 30 frames per second (fps) and the shutter speed to 1/1000 s considering the image analysis time. Table 1 shows the detailed specifications of the equipment used in the experiment.

### 3.3. Analysis of Measurement Results

The VDMS using UAV measurement was performed by capturing the images of the target using a UAV camera in flight and postprocessing them. The displacement was extracted by analyzing the measurement target and background target after setting the ROI in the video to reduce the analysis time, as shown in Figure 4.

The measurement results from the third cycle after the start of measurement to the last vibration were used as data for analysis. For error analysis, the data of all sensors were linearly interpolated, and the sampling rate was matched to 2000 Hz.

#### 3.3.1. Analysis of Displacement Response Measurement Results

The calibration method presented in Section 2 was used to estimate the displacement by the movement ($D_{3}$) and rotation ($D_{2}$) of the UAV to determine the absolute displacement of the measurement target. Figure 5 shows the result of the experiment with an excitation frequency of 2 Hz and an excitation displacement of 100 mm. Figure 5 shows that the movement of the UAV generates a displacement of up to 19.823 mm, whereas the rotation of the UAV generates a displacement of up to 1.368 mm.

Figure 6a,c show displacement response graphs for the experiment with an excitation frequency of 2 Hz and an excitation displacement of 10 mm and the experiment with an excitation frequency of 2 Hz and an excitation displacement of 100 mm. Furthermore, peaks were magnified in Figure 6b,d to examine the accuracy at the displacement peak. The VDMS using a UAV demonstrated a tendency similar to those of the LDS and LVDT, but a difference of 1–2 mm was observed at the peak.

The accuracy of displacement measurement was examined by obtaining the percentage errors of the RMS values of displacement with the LDS. Figure 7 shows the percentage errors of the RMS values of the LVDT and UAV when compared with the LDS. The LVDT’s difference in error as per the excitation frequency was not large; however, the error increased when the maximum excitation displacements were ~10 and 100 mm. The maximum error of the RMS value of the LVDT was 2.851% under the experiment with an excitation frequency of 1.4 Hz and an excitation displacement of 15 mm, whereas the minimum error was 0.009% under the experiment with an excitation frequency of 2.8 Hz and an excitation displacement of 60 mm. The average error was 0.541%. For the VDMS using a UAV, the maximum error of 4.589% occurred under the experiment with an excitation frequency of 3 Hz and an excitation displacement of 5 mm, and the minimum error of 0.003% was observed under the experiment with an excitation frequency of 0.8 Hz and an excitation displacement of 95 mm. The average error was 0.622%, which was similar to those of the LDS and LVDT; however, the error tended to decrease as the excitation displacement increased.

#### 3.3.2. Analysis of Measurement Results in the Frequency Domain

Figure 8 shows the responses of the experiment with an excitation frequency of 2 Hz and an excitation displacement of 10 mm and the experiment with an excitation frequency of 2 Hz and an excitation displacement of 100 mm in the frequency domain using the power spectral density (PSD) function. All of the displacement sensors clearly exhibited peaks in the frequency domain, and the same excitation frequency was observed in all the tests. However, the PSD amplitude showed a difference, as in the displacement response results.

The LVDT demonstrated a maximum error of 2.8929% in the experiment with an excitation frequency of 1.4 Hz and an excitation displacement of 15 mm and a minimum error of 0.0013% in the experiment with an excitation frequency of 2.2 Hz and an excitation displacement of 55 mm. The average error of the LVDT was 0.7697%, which was similar to that of the LDS, as shown in Figure 9. Furthermore, the VDMS using a UAV showed a maximum error of 4.0633% in the experiment with an excitation frequency of 0.4 Hz and an excitation displacement of 25 mm and a minimum error of 0.0082% in the experiment with an excitation frequency of 1 Hz and an excitation displacement of 65 mm. The average error of the VDMS using a UAV was 0.8164%, which was not significantly different from that of the LVDT.

## 4. Structure Measurement Accuracy Verification Test

### 4.1. Test Overview

Model structures with one and two stories were excited to verify the structure measurement accuracy of the VDMS using a UAV, and the accuracy was analyzed compared with the LDS utilized in Section 3. Accelerometers were used additionally for analyzing the accuracy of the identification of dynamic characteristics. The model structure was composed of aluminum alloy masses with dimensions of 600 mm × 600 mm × 20 mm and steel columns with a width of 50 mm. The thickness (3.0 mm) of the columns was significantly smaller compared with their width to induce the weak-axis direction behavior of the entire structure. For the accelerometers, KS48C from MMF GmbH [47] was used. The accelerometers and the LDS were installed on all floors. Figure 10 shows the front view of a two-story structure in an experiment to verify the structure measurement accuracy, where the targets were attached on the ground, first, and second floors. Similar to the vibration measurement performance verification test, UAV imaging was performed from the front to measure the measurement and background targets.

The structure was constructed in the range of one and two stories. The highest floor of the structure was excited, and free vibration was applied two times. The mode frequency, damping ratio, and mode shape, based on the displacement response of the structure, were compared with the LDS and accelerometer measurement results.

### 4.2. Analysis of Measurement Results

#### 4.2.1. Mode Frequency Estimation Results

Figure 11 and Figure 12 show the free vibration measurement results for the structures with one and two stories in the frequency domain using the PSD function. Only the 0 to 6 Hz frequency band clearly indicated the mode frequency peaks of the model structure. All of the results from the accelerators, LDS, and the VDMS using a UAV were the same for the mode frequency of the one- and two-story structures. In Figure 12a,b, harmonic peaks with a frequency of an integer multiple of the first mode frequency occurred around 3.8 Hz, and these harmonic peaks were ignored when estimating mode frequencies and mode shapes. Table 2 lists the mode frequency estimation results for each number of stories.

#### 4.2.2. Damping Ratio Estimation Results

The damping ratio was estimated using the logarithmic decrement method in the time domain. The damping ratio estimation results used the response from the highest floor of the structure, and the errors of LDS and the VDMS using a UAV were compared on the basis of the accelerometers.

The logarithmic decrement method represents the degree of reduction in amplitude at a logarithmic rate after one cycle in the time history measurement results of a free-vibrating structure. In this study, the damping ratio was estimated in 30 positive and negative peak sections in the measurement results, and the average values were derived, as listed in Table 3.

The damping ratios of the accelerometer on the highest floor of the two-story structure were determined as 0.5890% in the positive section and 0.6877% in the negative section, and the average was estimated as 0.6383%. Meanwhile, the damping ratios of the LDS were 0.6257% in the positive section and 0.6198% in the negative section, and the average was estimated as 0.5338%. For the VDMS using a UAV, the damping ratios were 0.6278% in the positive section and 0.7247% in the negative section, and the average was determined as 0.6368%.

In the one-story structure, the error of the LDS was 14.4866% with the accelerometer, whereas the error of the VDMS using a UAV was 2.0993%. In the two-story structure, the error of LDS was 2.4396%, whereas the error of the VDMS using a UAV was 0.2428%. When the estimated damping ratio of the VDMS using a UAV was compared with that of the accelerometer, there was no significant difference from the LDS results.

#### 4.2.3. Mode Shape Estimation Results

The mode shape was expressed using the frequency domain decomposition (FDD) method [48]. Specifically, the FDD method obtains the natural frequency and mode shape by applying singular value decomposition to the result from the PSD function. It can obtain the mode shape even when the characteristics of the excitation load are unknown. Figure 13 shows the mode shapes of the first and second modes obtained from the two-story structure measurement results of the accelerometers, LDS, and the VDMS using a UAV.

The modal assurance criterion (MAC) values of all specimens for each mode were obtained from the measurements of LDS and the VDMS using a UAV, and the average of the mode MAC values was obtained for each number of stories, as listed in Table 4.

For LDS and the VDMS using a UAV, the lowest MAC values (0.9900 and 0.9972, respectively) were observed in the second mode of the two-story structure. Furthermore, LDS and the VDMS using a UAV clearly showed mode shapes and relatively high MAC values in all tests.

## 5. Discussion and Conclusions

In this study, vibrations were generated using a shaking table by increasing the excitation frequency by 0.2 Hz (from 0.2 to 3 Hz) and the displacement by 5 mm (from 5 to 100 mm) to verify the dynamic measurement accuracy of the VDMS using a UAV. Specifically, various experiments were conducted using the UAV, and the measurement results were compared with those of LDS and LVDT. Additionally, free vibration was applied to the multi-DOF model structure, and measurements were performed using LDS and accelerometers to verify the performance for identifying structural dynamic characteristics.

In the test for vibration measurement performance verification, the vibration measurement accuracy of the VDMS using a UAV was verified by calculating the RMS of the measurements of the LVDT and the VDMS using a UAV based on the LDS measurements. In the RMS percentage error analysis results, the LVDT showed no error tendency according to the test case and exhibited a maximum error of 2.851%. The LVDT showed such type of tendency because the peak could not be accurately achieved as the excitation displacement and excitation frequency increased because of the limit of the contact sensor. For the VDMS using a UAV, the error tended to decrease as the excitation displacement increased, and the maximum error was 4.589%. The reason why the error of the VDMS using a UAV was larger than that of the LVDT was that the movement of the UAV could not be completely eliminated because of the distortion by the camera lens and the rotation angle estimation error in the course of subtracting the displacement value of the background target from that of the measurement target. In addition, regardless of the frequency, a relatively large error appeared in the displacement measurement of 10 mm or less, and the reliability of the VDMS using the UAV was lowered in this section.

In the structure measurement accuracy verification test, the mode frequency, damping ratio, and mode shape estimation results were compared. In the mode frequency estimation results, all of the accelerators, the LDS, and the VDMS using a UAV exhibited the same results. In the estimation of the damping ratio, all measurement sensors showed similar results, excluding the damping ratio of the two-story structure of the LDS. In the estimation of the mode shape, all measurement sensors clearly showed mode shapes and high MAC values.

Given the above results, the VDMS using a UAV can replace conventional displacement sensors. Additionally, more accurate measurements would be possible if displacement is measured by applying a filter that can remove the residual tendency due to the movement of the drone. Moreover, methods for removing the movement of UAVs by finding key points that can correct such movements on the same plane as the measurement target must be studied.

## Figures and Tables

**Figure 1 sensors-23-03232-f001:**
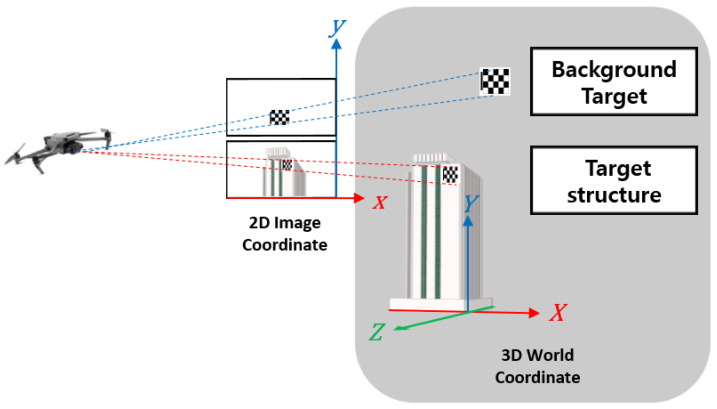
Configuration of a point in a 2D image coordinate and the corresponding 3D world coordinate.

**Figure 2 sensors-23-03232-f002:**
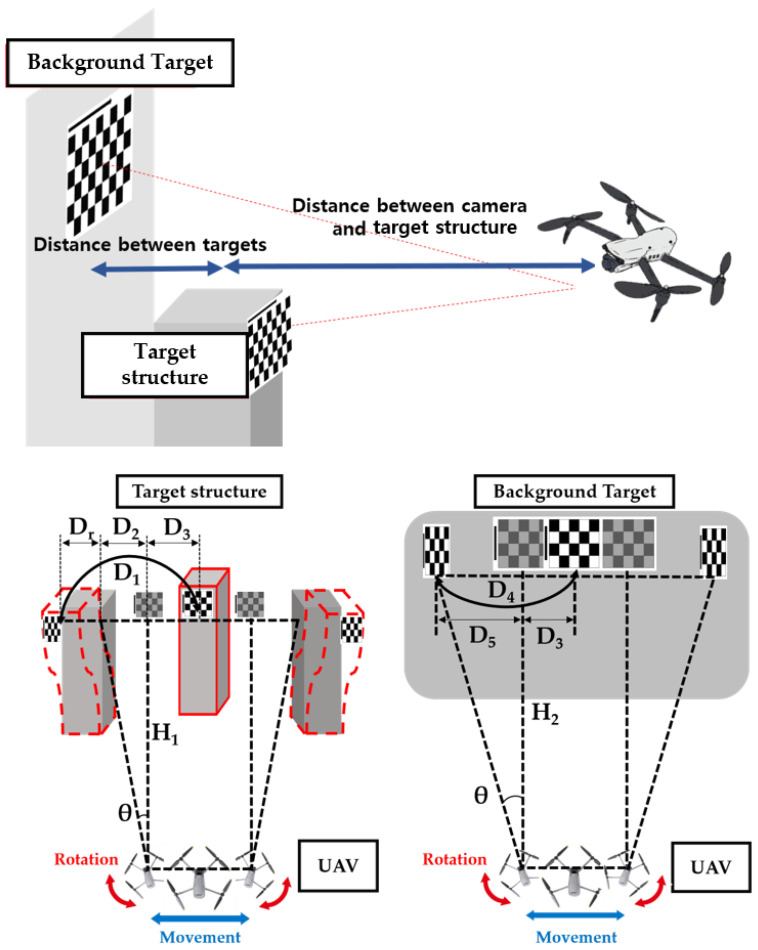
Overview of a UAV’s motion elimination method.

**Figure 3 sensors-23-03232-f003:**
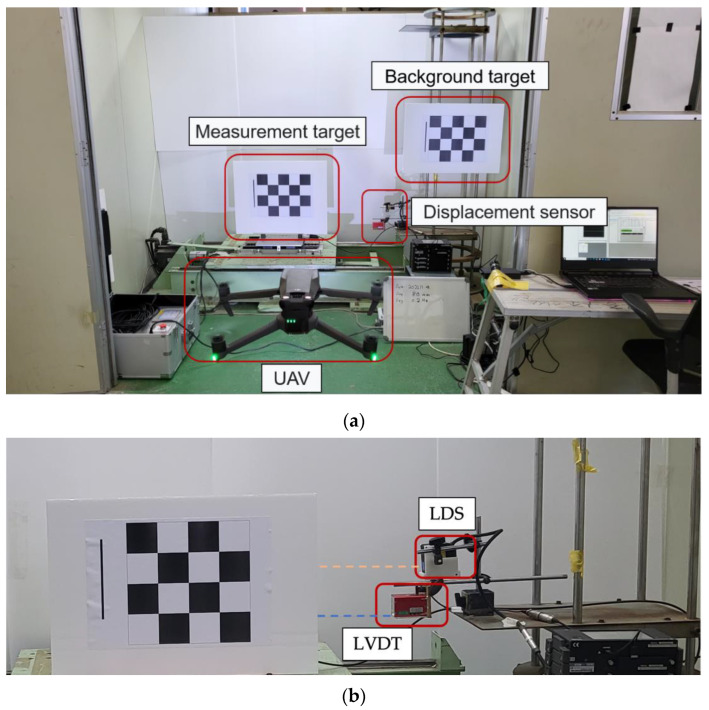
Front view and sensor setting status of the test for the vibration measurement performance verification: (**a**) front view of the test; (**b**) enlarged LDS and LVDT installation view and orange and blue dashed lines indicate the measurement direction of the displacement sensor.

**Figure 4 sensors-23-03232-f004:**
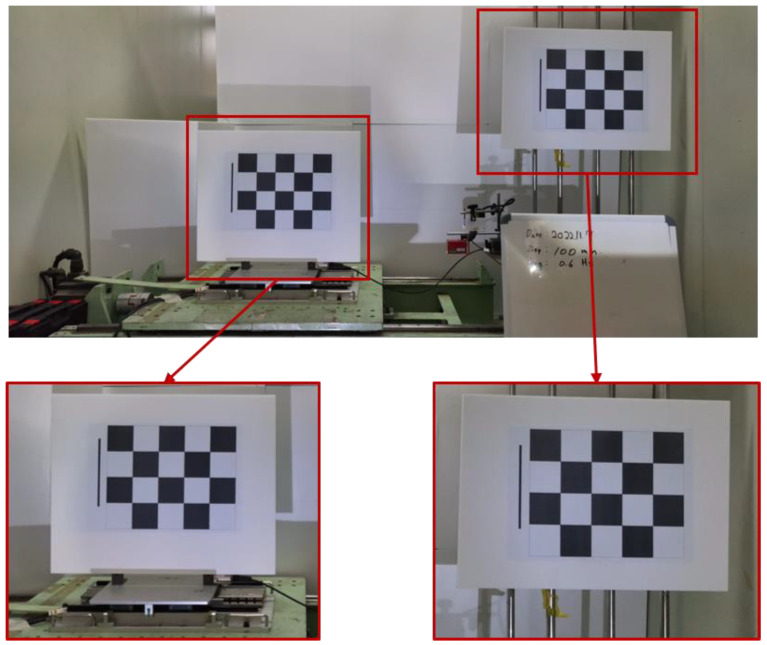
Experiment view showing specific ROI in UAV images (the lower left image represents the measurement target, and the lower right image represents the background target).

**Figure 5 sensors-23-03232-f005:**
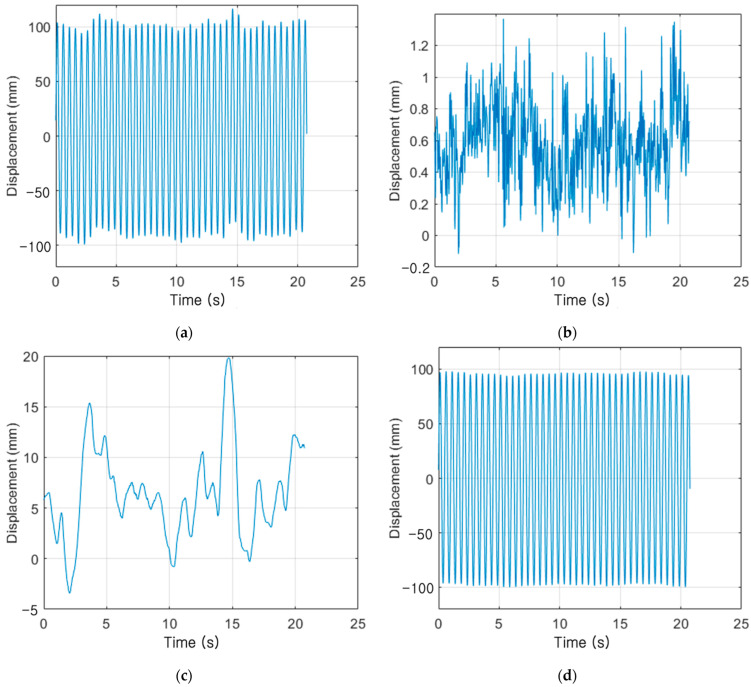
VDMS using the UAV result of the experiment with an excitation frequency of 2 Hz and an excitation displacement of 100 mm: (**a**) total displacement of the measurement target captured by the moving and rotating UAV ($D_{1}$); (**b**) displacement generated by the UAV’s rotation ($D_{2}$); (**c**) displacement generated by the UAV’s movement ($D_{3}$); and (**d**) actual displacement generated in the measurement target ($D_{r}$).

**Figure 6 sensors-23-03232-f006:**
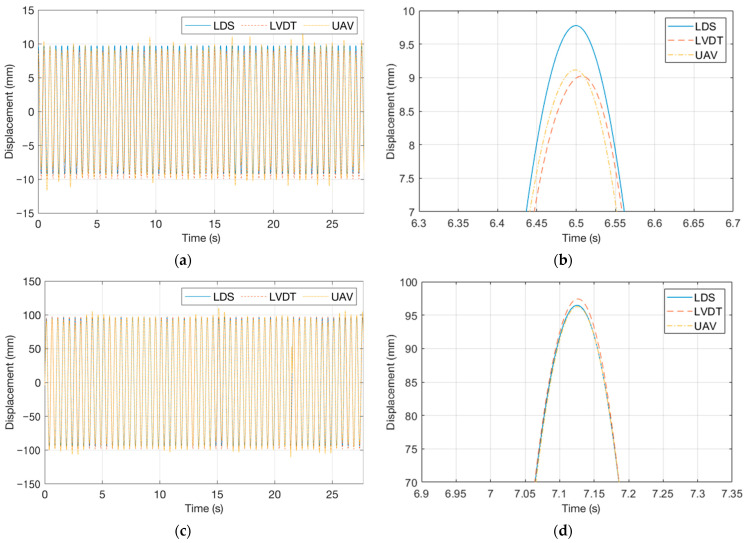
Displacement response graphs of the vibration measurement performance verification test: (**a**) response for the experiment with an excitation frequency of 2 Hz and an excitation displacement of 10 mm; (**b**) one peak in (**a**); (**c**) response for the experiment with an excitation frequency of 2 Hz and an excitation displacement of 100 mm; and (**d**) one peak in (**c**).

**Figure 7 sensors-23-03232-f007:**
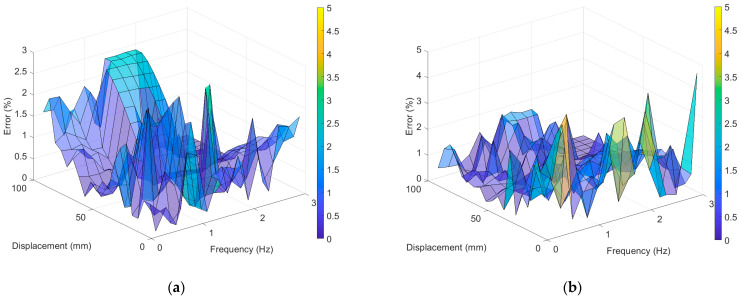
Percent error of the RMS value compared with LDS: (**a**) LVDT error and (**b**) error for the VDMS using a UAV.

**Figure 8 sensors-23-03232-f008:**
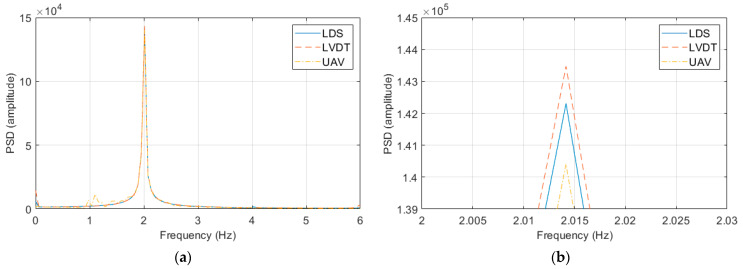
Frequency response graphs of the vibration measurement performance verification experiment: (**a**) experimental result at an excitation frequency of 2 Hz and an excitation displacement of 10 mm; (**b**) peak interval graph of (**a**); (**c**) experimental result at an excitation frequency of 2 Hz and an excitation displacement of 100 mm; and (**d**) peak interval graph of (**c**).

**Figure 9 sensors-23-03232-f009:**
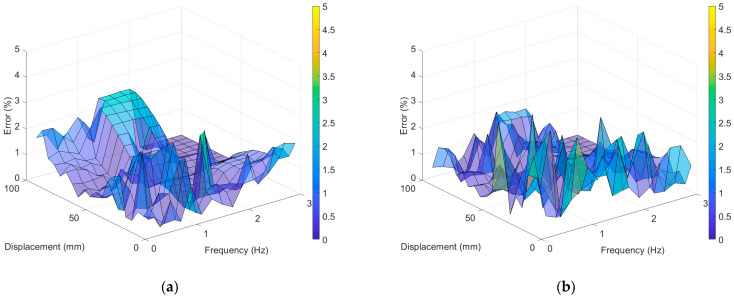
PSD amplitude error compared with LDS: (**a**) LVDT error; (**b**) error of the VDMS using a UAV.

**Figure 10 sensors-23-03232-f010:**
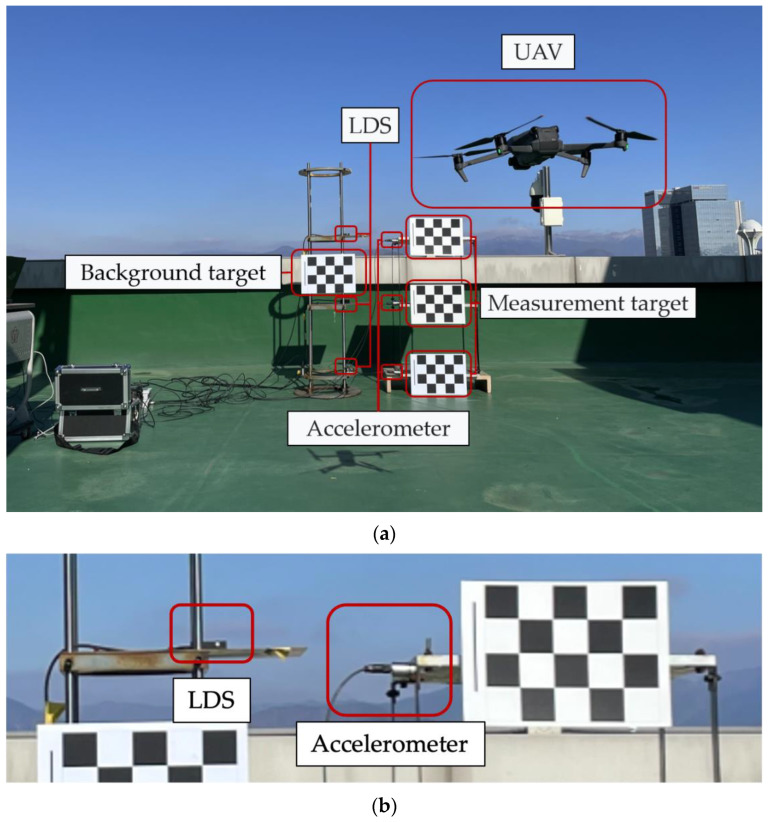
Front view of a two-story structure in an experiment to verify the structure measurement accuracy: (**a**) front view of a two-story structure; (**b**) enlarged 2nd floor accelerometer and LDS installation view.

**Figure 11 sensors-23-03232-f011:**
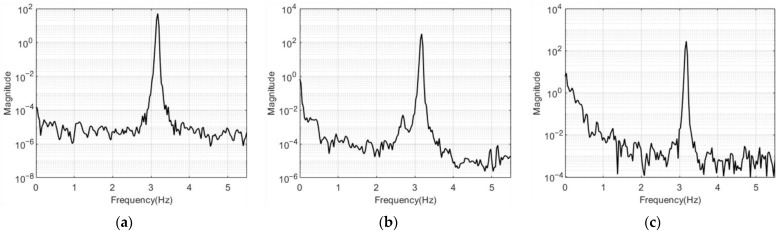
Frequency response graphs for the one-story model structure: (**a**) acceleration response; (**b**) LDS response; (**c**) response of the VDMS using a UAV.

**Figure 12 sensors-23-03232-f012:**
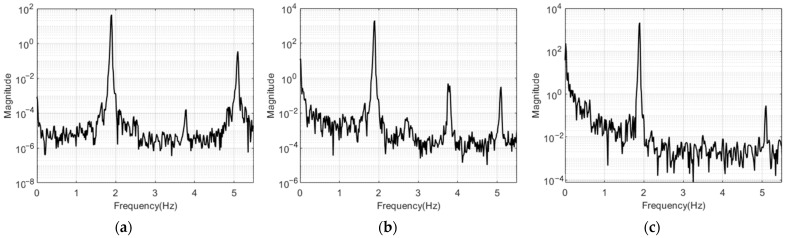
Frequency response graphs for the two-story model structure: (**a**) acceleration response; (**b**) LDS response; (**c**) response of the VDMS using a UAV.

**Figure 13 sensors-23-03232-f013:**
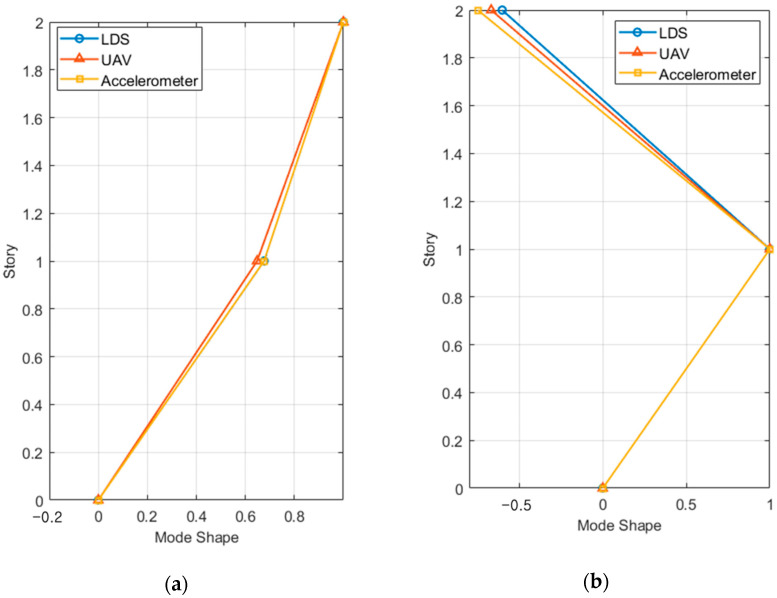
Mode shapes of the two-story model structure: (**a**) first mode and (**b**) second mode.

**Table 1 sensors-23-03232-t001:** Vibration measurement performance verification test equipment.

Component	Model	Technical Specifications
Shaking table	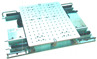 TE Solution’s portable 1-axis vibration device	Maximum speed: 4.8 m/sRated thrust: 140 NRated current: 2.2 ArmsThermal resistance: 1.48 K/W
LVDT	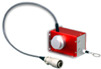 Tokyo Sokki’s DP-500E	Capacity: 500 mmSensibility: 20 μ strain/mmNonlinearity: 0.3% rated output
LDS	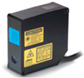 KAIS’s KL3-W400	Measurement distance: 500 mmMeasurable range: ±200 mmOptical mode: Diffuse reflectionResolution: 10 μm
Data logger	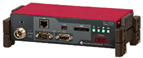 Tokyo Sokki’s TMR-311	Sampling time: 1–1000 ms (set by every 1 ms)Power supply: AC 100–240 V, 50/60 Hz, 100 VA at maximum Measurement range: ±20,000 × 10^−6^ strain (bridge excitation DC 2V)
UAV	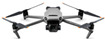 DJI’s MAVIC 3	Takeoff weight: 895 gMax flight time: 46 minMax hovering time: 40 minMax wind speed resistance: 12 m/s
UAV Camera	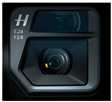 DJI’s Hasselblad Camera	Sensor: 4/3 CMOS, Effective pixels: 20 MPISO Range: 100–6400Shutter Speed: 8–1/8000 sFOV: 84°Format Equivalent: 24 mmAperture: f/2.8 to f/11Focus: 1 m to ∞ (with autofocus)Video Resolution: H.264/H.2655.1K: 5120 × 2700@24/25/30/48/50fpsDCI 4K: 4096 × 2160@24/25/30/48/50/60/120fps4K: 3840 × 2160@24/25/30/48/50/60/120fpsFHD: 1920 × 1080@24/25/30/48/50/60/120/200fps

**Table 2 sensors-23-03232-t002:** Mode frequency estimation results for one and two-story structures (Hz).

Number of Stories	Mode	Accelerometer	LDS	UAV
1	1st	3.17383	3.17383	3.17383
2	1st	1.89209	1.89209	1.89209
	2nd	5.09644	5.09644	5.09644

**Table 3 sensors-23-03232-t003:** Result of damping ratio estimation on the top floor of one- and two-story structures.

Number of Stories	Accelerometer	LDS	UAV
1	0.447102%	0.382332% (14.48663)	0.456488% (2.09930)
2	0.638349%	0.622776% (2.43957)	0.636799% (0.24281)

() Percentage error with an accelerometer.

**Table 4 sensors-23-03232-t004:** Mode MAC value estimation results for one- and two-story structures.

Number of Stories	Number of Modes	LDS	UAV
1	1st	1	1
2	1st	0.9999	0.9996
	2nd	0.9900	0.9972

## Data Availability

Data available on request due to restrictions.

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
