# Peer review of "Reliability Assessment of a Vision-Based Dynamic Displacement Measurement System Using an Unmanned Aerial Vehicle"

_sensors, 2023, doi:10.3390/s23063232_

Round 1

Reviewer 1 Report

Dear authors,

Below please find some remarks considering your paper.

·         In the case of the title and the abstract, those elements must be understandable to none familiar with the field reader. It is advised to use no abbreviations if possible. In the abstract, the authors are using abbreviations, which should be avoided in the abstract and those abbreviations should be introduced for the first time in the main text. Otherwise, the abstract is good.

·         The reference list is good with some minor adjustments to be made in case of missing state-of-the-art elements – look next point.

·         The introduction part is written clearly. However:

o   The authors in a very nice manner introduce the topic of contact and non-contact measurements. However, the evaluation of none contact methods is very short and not presenting the full state-of-the-art alternatives, especially optical systems (lines 46-54). Some possibilities for other optical systems that can measure the displacement of even very big objects (buildings) include video recorders and virtual visual sensors (e.g. https://doi.org/10.1177/14759217145228), a digital image correlation (DIC) (e.g https://doi.org/10.1016/j.engstruct.2020.110551) and especially 3D Lase Doppler Vibrometry (e.g DOI: 10.3390/s23031263 ). In this last case, this is an optical system dedicated to vibration measurements but also possible to measure displacement and is used for quality control also in the case of civil engineering structures or building structural elements. Additionally, the system apart from displacement allows the measurement vibration (natural frequencies, mode shapes and damping) without the need to use accelerometers and another acquisition system. Please make the correction and the beginning of the introduction. Without it, there is no proper background to the research you are presenting. Especially, since the authors present an alternative solution with the use of UAV, which reliability is in question.

o   The last paragraph where the aim, goal and especially the novelty of the study is to be presented is to be rewritten. These elements are presented but novelty is not clearly presented, especially, since the authors are writing that there are many studies on the topic of the use of UAVs for structural monitoring.

·         Material and methods, results:

o   The authors are presenting their experimental setup, the model of the accelerometer used is presented but without its parameters. Additionally, was it a uni-axial o 3-axial accelerometer? What was the orientation? Could not find the information, with accurate placements. This information is not also given in Fig. 12.

o   Fig. 10 is extremally important but details are hardly visible, especially in the case of sensor mounting and orientation. Maybe add zoom-in photos of the presented elements.

·         The conclusion and discussion are good. Additionally, please present stronger the novelty of the paper.

In conclusion: The paper is clear. The biggest advantage is the presence of both tests in laboratory conditions and in the case of real-life applications. The paper needs however a better presentation of novelty and especially adding some alternative techniques in the introduction.

Due to some small flaws, the reviewer is marking the paper for minor revisions. Hope the authors will use some suggestions to improve this otherwise very interesting and good, in case of results presentation, paper.  

Author Response

Thank you for your review comments.

o   The authors in a very nice manner introduce the topic of contact and non-contact measurements. However, the evaluation of none contact methods is very short and not presenting the full state-of-the-art alternatives, especially optical systems (lines 46-54). Some possibilities for other optical systems that can measure the displacement of even very big objects (buildings) include video recorders and virtual visual sensors (e.g. https://doi.org/10.1177/14759217145228), a digital image correlation (DIC) (e.g https://doi.org/10.1016/j.engstruct.2020.110551) and especially 3D Lase Doppler Vibrometry (e.g DOI: 10.3390/s23031263 ). In this last case, this is an optical system dedicated to vibration measurements but also possible to measure displacement and is used for quality control also in the case of civil engineering structures or building structural elements. Additionally, the system apart from displacement allows the measurement vibration (natural frequencies, mode shapes and damping) without the need to use accelerometers and another acquisition system. Please make the correction and the beginning of the introduction. Without it, there is no proper background to the research you are presenting. Especially, since the authors present an alternative solution with the use of UAV, which reliability is in question.

  • In the introduction, we have added an optical system, video recorder and virtual vision sensor, as well as digital image correlation and 3D laser Doppler vibrometer as newer alternatives.

o   The last paragraph where the aim, goal and especially the novelty of the study is to be presented is to be rewritten. These elements are presented but novelty is not clearly presented, especially, since the authors are writing that there are many studies on the topic of the use of UAVs for structural monitoring.

  • In the last paragraph of the introduction, the difference with the existing literature was added and corrected.

o   The authors are presenting their experimental setup, the model of the accelerometer used is presented but without its parameters. Additionally, was it a uni-axial o 3-axial accelerometer? What was the orientation? Could not find the information, with accurate placements. This information is not also given in Fig. 12.

  • A 1-axis accelerometer was used for the accelerometer, and the expression was modified to 1-axis accelerometer in the text. Figure 12 also shows the placement of the accelerometers and displacement sensors together.

o   Fig. 10 is extremally important but details are hardly visible, especially in the case of sensor mounting and orientation. Maybe add zoom-in photos of the presented elements

  • We have added an zoom-in photo to Figure 10.

Reviewer 2 Report

This manuscript validates the dynamic measurement of a vision-based displacement measurement system (VDMS) using unmanned aerial vehicles (UAVs) and compares the measurement results with those of LDS and LVDT. The study also analyzes the problem of camera lens distortion in the visual method, which leads to measurement errors larger than those of the linear variable differential transformer (LVDT). Overall, this research experimentally verifies the non-contact measurement method based on vision and UAV and points out its advantages and disadvantages in technical applications.

(1) At the thematic level, this study compares several representative displacement measurement methods, which is beneficial for promoting new methods. However, the description of the effective measurement range of VDMS in the final summary is not clear enough.

(1) The introduction section could benefit from a more comprehensive overview of the problems related to computer vision detection and references to relevant research (Seismic performance evaluation of recycled aggregate concrete-filled steel tubular columns with field strain detected via a novel mark-free vision method. Structures, 2022).

(2) The document includes a total of 35 references, of which only 12 were published in the last 5 years. Therefore, the total number is insufficient, and their actuality is not high.

(3) It is recommended that the author conduct further research on the shortcomings of VDMS analyzed in this experiment. The current work is only for engineering applications.

(4) The abstract is complete, well-structured, and effectively explains the contents of the document. However, it is suggested that the abstract clearly state the limited measurement range of this technology and numerically demonstrate the measurement errors of various measurement methods. The keywords should also include "reliability assessment."

(6) In the introduction, the author summarizes several measurement methods but does not present a comprehensive evaluation.

(7) Vision technology applications in SHM should also be introduced for a full glance at the scope of related areas. For crack classification, references keywords such as integrated generative adversarial networks and improved VGG model, should be mentioned. For crack width detection, references keywords such as backbone double-scale features for improved detection automation, should be mentioned.

(8) In the method section, it is necessary to explain how to consider the position of the calibration board and the algorithm used for real-time identification of markers.

(9) Chapters 3 and 4 provide detailed descriptions of the experimental equipment and outdoor experimental setup. The experimental settings are reasonable, the equipment is complete, and the experimental results are credible.

(10) In the final section, the author does not clearly indicate the reliable measurement range of VDMS in this field.

(11) Has the author considered the limitation of the field of view of the UAV vision system, especially when used in large buildings?

Author Response

Thank you for your careful reading.

  • At the thematic level, this study compares several representative displacement measurement methods, which is beneficial for promoting new methods. However, the description of the effective measurement range of VDMS in the final summary is not clear enough.

  • We have added information about the reliable measurement range of VDMS using UAVs according to the test results.

  • The introduction section could benefit from a more comprehensive overview of the problems related to computer visiondetection and references to relevant research (Seismic performance evaluation of recycled aggregate concrete-filled steel tubular columns with field strain detected via a novel mark-free vision method. Structures, 2022).

  • We agree with you. Therefore, we have added a comprehensive overview of computer vision detection and related research to the introduction.

  • The document includes a total of 35 references, of which only 12 were published in the last 5 years. Therefore, the total number is insufficient, and their actuality is not high.

  • By adding a comprehensive overview in (2), the number of references and references to recent studies has increased.

  • It is recommended that the author conduct further research on the shortcomings of VDMS analyzed in this experiment. The current work is only for engineering applications.

  • Thank you for your comments. So in next study, I will develop a VDMS using an unmanned vehicle that does not require a background target.

  • The abstract is complete, well-structured, and effectively explains the contents of the document.However, it is suggested that the abstract clearly state the limited measurement range of this technology and numerically demonstrate the measurement errors of various measurement methods. The keywords should also include "reliability assessment."

  • The measurement range was described in the abstract, and “reliability assessment” was added to the keywords.

  • In the introduction, the author summarizes several measurement methods but does not present a comprehensive evaluation.

  • The manuscript has been updated to include comprehensive research in response to feedback from all reviewers. The advantages and disadvantages of each piece of research have been thoroughly presented and discussed in the manuscript.

  • Vision technology applications in SHMshould also be introduced for a full glance at the scope of related areas. For crack classification, references keywords such as integrated generative adversarial networks and improved VGG model, should be mentioned. For crack width detection, references keywords such as backbone double-scale features for improved detection automation, should be mentioned.

  • We have included crack classification and crack width detection in a comprehensive overview and related areas in the introduction.

  • In the method section, it is necessary to explain how to consider the position of the calibration board and the algorithm used for real-time identification of markers.

  • In the method section, we have added the position of the calibration board and the algorithm used.

  • Chapters 3 and 4 provide detailed descriptions ofthe experimental equipment and outdoor experimental setup. The experimental settings are reasonable, the equipment is complete, and the experimental results are credible.

  • Thank you for your review comment.

  • In the final section, the author does not clearly indicate the reliable measurement range of VDMS in this field.

  • We have added information about the reliable measurement range of VDMS using UAVs according to the test results.

  • Has the author considered the limitation of the field of view of the UAV vision system, especially when used in large buildings?

  • In this study, we verified reliability in the most general use. Therefore, field conditions were not considered. However, we will do field tests together in future studies and will apply your comments.

Reviewer 3 Report

The manuscript is devoted to testing a vision-based displacement measurement system using an unmanned aerial vehicle. 

The authors showed the relevance of their research, the differences of this method from others.

The manuscript presents a large set of tests of the vision-based displacement measurement system (VDMS).

There are the following questions and comments on the text of the Manuscript:

1. Section 3.1: what is the rationale for choosing a frequency from 0.2 to 3 Hz and an offset of 5-100 mm?

2. Section 3.2 does not show the optical and lighting parameters of the UAV camera.

3. Figures 6 (a, c) are uninformative.

4. The Introduction talks about the construction of multi-storey buildings, while in Section 4 tests were carried out only on two-storey structures.

5. Tables 2 and 3 do not specify units of measurement (Hz?). Where does the accuracy of 6-7 significant digits come from in the results of the tables?

6. Section 5 should be called Discussion and Conclusions or Conclusions.

Author Response

Thank you for your careful reading.

There are the following questions and comments on the text of the Manuscript:

  1. Section 3.1: what is the rationale for choosing a frequency from 0.2 to 3 Hz and an offset of 5-100 mm?

  • For the frequency, the sampling rate of VDMS using an unmanned body is 30Hz, and the interval up to 3Hz was set considering the Nyquist frequency. The displacement is set to the maximum according to the limit of the shaking table.

  1. Section 3.2 does not show the optical and lighting parameters of the UAV camera.

  • I agree that the answer to your question should be included in the text. So we added the optical and lighting parameters of the UAV camera.

  1. Figures 6 (a, c) are uninformative.

  • Information of Figures 6 (a, c) have been added.

  1. The Introduction talks about the construction of multi-storey buildings, while in Section 4 tests were carried out only on two-storey structures.

  • In this study, the model structure test measured the free vibration response. The sampling frequency of VDMS using UAV was 30Hz, so it was impossible to identify higher order modes in structures with 3 or more storeyes.

  1. Tables 2 and 3 do not specify units of measurement (Hz?). Where does the accuracy of 6-7 significant digits come from in the results of the tables?

  • We have corrected Tables 2 and 3 to indicate units. Table 2 is in Hz and Table 3 is in %.

  1. Section 5 should be called Discussion and Conclusions or Conclusions.

  • I agree with your comments. We have corrected Section 5 to “Discussion and Conclusions”.